



# Climatological assessment of the vertically resolved optical and microphysical aerosol properties by lidar measurements, sunphotometer, and in-situ observations over 17 years at UPC Barcelona

Simone Lolli[1,2], Michaël Sicard[2,*], Francesco Amato[1], Adolfo Comeron[2], Cristina Gíl-Diaz[2], Tony C. Landi[3], Constantino Munoz-Porcar[2], Daniel Oliveira[2], Federico Dios Otin[2], Francesc Rocadenbosch[2,4], Alejandro Rodriguez-Gomez[2], Andrés Alastuey[5], Xavier Querol[5], and Cristina Reche[5]

[1]Italian National Research Council CNR-IMAA, 85050 Tito Scalo (PZ), Italy
[2]CommSensLab, Dept. of Signal Theory and Communications, UPC, 08034 Barcelona, Spain
[3]CNR-ISAC, Via Gobetti 101, 40129, Bologna, Italy
[4]Institut d'Estudis Espacials de Catalunya (Institute of Space Studies of Catalonia, IEEC), Barcelona, Spain, E-08034
[5]Institute of Environmental Assessment and Water Research (IDAEA-CSIC), Barcelona, Spain
[*]Now at Laboratoire de l'Atmosphère et des Cyclones, Université de La Réunion, 97744, Saint Denis, France

**Correspondence:** Simone Lolli (simone.lolli@upc.edu)

**Abstract.** Aerosols are one of the most important pollutants in the atmosphere and have been monitored for the past few decades by both remote sensing and in situ observation platforms to assess the effectiveness of government-managed reduction emission policies and assess their impact on the radiative budget of the Earth's atmosphere. In fact, aerosols can directly modulate incoming short-wave solar radiation and outgoing long-wave radiation and indirectly influence cloud formation, lifetime,
and precipitation. In this study, we quantitatively evaluated long-term temporal trends and seasonal variability from a climatological point of view of the optical and microphysical properties of atmospheric particulate matter at the Universitat Politècnica de Catalunya, Barcelona, Spain, over the past 17 years, through a synergy of lidar, sunphotometer, and in situ concentration measurements. Interannual temporal changes in aerosol optical and microphysical properties are evaluated through the seasonal Mann-Kendall test. Long-term trends in the optical depth of the recovered aerosol, the Ångström exponent (AE) and the
concentrations of $PM_{10}$, $PM_{2.5}$ and $PM_1$ reveal that emission reduction policies implemented in the last decades were effective in improving air quality, with consistent drops in PM concentrations and optical depth of aerosols. The seasonal analysis of the 17-year average vertically resolved aerosol profiles obtained from lidar observations shows that during summer the aerosol layer can be found up to an altitude of 5 km, after a sharp decay in the first km. In contrast, during the other seasons, the backscatter profiles fit a pronounced exponential decay well with a well-defined scale height. Long-range transport, especially
dust outbreaks from the Sahara Desert, is likely to occur throughout the year. During winter, the dust aerosol layers are floating above the boundary layer, while during the other seasons they can penetrate the layer. This study sheds some light on meteorological processes and conditions that can lead to haze formation and helps decision makers adopt mitigation strategies to preserve large metropolitan areas in the Mediterranean basin.



## 1   Introduction

Aerosols are small particles suspended in the air and can have a significant impact on both human health and the environment. These short-lived forcers have a brief atmospheric lifetime, typically a few days to a few weeks, and they play a crucial role in shaping the Earth's climate and air quality. These particles come from a variety of sources, including natural processes such as dust and sea salt and human activities such as burning fossil fuels, industrial processes, and agriculture.

The effect of aerosols on human health is mainly through inhalation. Particles can cause respiratory problems, such as asthma and bronchitis, and have been linked to increased rates of heart disease and stroke. Long-term exposure to aerosols has also been associated with lung cancer and other respiratory diseases. The elderly and children, who are more susceptible to respiratory problems, are particularly vulnerable to the negative effects of aerosols (Pope III and Dockery, 2006).

Aerosols also have a profound impact on climate because they can directly scatter and absorb solar radiation, influencing the energy balance of the Earth and causing cooling or warming (Crosier et al., 2007; Landi et al., 2021). Natural and anthropogenic emissions have different effects on incoming short-wave (SW) solar radiation and outgoing long-wave (LW) radiation emitted by the Earth's surface. Some aerosols, such as sulfates and marine aerosols, reflect short-wave radiation and cool the surface by increasing the planetary albedo. However, the black carbon aerosol layers absorb the incoming SW radiation, warming the air around them during the day. Dust aerosols can affect both incoming SW radiation and outgoing LW radiation, although to a lesser extent. The cooling effect (at surface) of aerosols is significant in regions with high levels of particulate pollution, such as South Asia and East Asia (Bilal et al., 2019; Yang et al., 2020; Huang et al., 2021) and certain areas of the United States (Tosca et al., 2017). The warming effect is most pronounced in the Arctic, where the concentration of aerosols is relatively low.

The presence of aerosols in the atmosphere also has an impact on atmospheric stability, vertical movements, and large-scale circulation. They can also affect regional hydrological cycles and cause significant regional climate sensitivity. In recent decades, human industrial activities have significantly altered the net aerosol radiative effect (RE) at the surface, particularly due to increased anthropogenic emissions.

The Fifth Assessment Report (AR5) of the Intergovernmental Panel on Climate Change (IPCC;Stocker (2014)) highlighted that aerosols have a negative effective radiative forcing (ERF) of -1.3 $W/m^2$ with respect to the industrial era (1750-2014). The effective radiative forcing due to aerosol-cloud interactions is the largest contributor to total aerosol ERF, -1.0 $W/m^2$ (medium confidence level), while the rest is due to aerosol-radiation interactions, -0.3 $W/m^2$. Regarding the Fifth Assessment Report, the estimate of the total aerosol ERF has improved with a higher magnitude and reduced uncertainty. This is supported by advances in understanding, modeling, and observational analysis. The new Sixth Assessment Report (AR6; (Adler et al., 2022)) undoubtedly shows that the total aerosol ERF is negative. Compared with previous assessments, the ERF due to aerosol-cloud interactions has increased, but the ERF due to aerosol-radiation interactions has decreased. The total aerosol ERF of 1750-2019 is less certain with a magnitude of -1.1 $W/m^2$, mainly due to recent changes in emissions. For this reason, it is of paramount importance to quantitatively assess how the optical and microphysical properties changed in the past decades.



In this study, we quantitatively evaluated changes in aerosol optical and microphysical properties by combining long-term data (2004-2020) in situ PM10, PM2.5 and PM1 concentrations, aerosol optical depth, and Ångström exponent AE obtained from NASA AERONET (AErosol RObotic NETwork; (Tanré et al., 1998)) sunphotometer network and lidar measurements obtained over the past 17 years at the Universitat Politecnica de Catalunya (UPC) in Barcelona, Spain. These measurements were obtained using a lidar deployed as part of the permanent observational network of EARLINET (European Aerosol Research Lidar Network; (Bösenberg and Matthias, 2003; Pappalardo et al., 2014)), which is dedicated to the study of atmospheric aerosols and clouds as part of the ACTRIS (Aerosols, Clouds, and Trace gases Research Infrastructure), which aims, in fact, to understand the properties and behavior of aerosols and clouds in the atmosphere of the Earth and to use this information to improve air quality and climate models.

Barcelona is one of the main industrialized metropolitan areas of the Mediterranean: a coastal city where air pollution is also exacerbated by the pollution of sea vessels (Mueller et al., 2011). In addition, Catalunya is in the middle of a complex region such as the Mediterranean, where the climate changes faster than in other parts of the world. In fact, according to IPCC AR6 (Ali et al., 2022) and other studies (Nicholls et al., 2008; Lewis et al., 2019), temperatures in the Mediterranean are increasing at a rate of 20% faster than the global average. The region's temperatures have already risen to 1.5 ° C higher than preindustrial levels. On the contrary, the global temperature increase has been slower, reaching approximately 1.1-1.3°C. If human greenhouse gas emissions were to stop immediately, the Mediterranean region is expected to experience a temperature increase of 2-4 ° C above 19th century levels by 2100. According to (Cramer et al., 2018), the evidence is clear: the already dry Mediterranean regions are becoming even drier. This is a milestone that the IPCC predicts the Mediterranean region will reach by the end of the century or even sooner. This analysis suggests that climate change at this level will transform large areas of southern Spain into deserts. Scrubber vegetation will replace deciduous forests, which will then move upward, replacing alpine conifer-based ecosystems. For all the previous reasons, quantifying changes in aerosol loading during the last 17 years can help shed some light on how these short-lived tracers can offset climate changes and help implement future adaptation strategies.

## 2 Materials and Methods

### 2.1 Instruments

a) *Lidar*

Light detection and ranging (Lidar) is an optically active remote sensing technique that has been increasingly used in atmospheric studies over the past few decades. Lidar measurements can provide valuable information on the vertical distribution of aerosol, cloud, and gas properties, including their optical and microphysical properties. Lidar measurements have been used in a variety of atmospheric studies, including monitoring air quality (Lolli et al., 2008, 2019), studying the vertical structure of clouds (Campbell et al., 2016, 2018; Lolli et al., 2017), and measuring greenhouse gases such as carbon dioxide (Abshire et al., 2010) and methane (Ehret et al., 2017). The very high spatial and temporal resolution of lidar measurements makes them particularly valuable for studying atmospheric processes on regional and local scales. Various lidar techniques, such as





elastic lidar, Raman lidar (Ansmann et al., 1990; Whiteman, 2003), dial lidar (Browell et al., 1998), Doppler lidar (Lolli et al.,
2013), and high-spectral-resolution laser (HSRL; Grund and Eloranta (1991)), make different assumptions, allowing retrieval
of optical and microphysical properties with varying degrees of accuracy.

Lidar observations are available within the framework of the EARLINET/ACTRIS project, as stated in Section 2.1.1. The
lidar instrument, deployed at UPC and part of the EARLINET/ACTRIS project since 2000, is a multiwavelength Raman lidar
using a Nd:Yag laser source, which is much more suitable than other sources such as $CO_2$ (Ciofini et al., 2003).

The UPC multispectral lidar system, which has eight channels, is the third lidar station conceptually designed at the Remote
Sensing Laboratory since the beginning of lidar research in 1993. The station is classified as a $3\beta+2\alpha+1WV+2\delta$ elastic/Raman
aerosol/water-vapor system. The laser source is a pulsed Nd:YAG (350 mJ per pulse) that emits simultaneously at the 1064-nm
(near-infrared, NIR) and 532-nm (vis) wavelengths. The system has 8 channels:

– 3 elastic channels (355, 532- and 1064-nm wavelength)

– 3 Raman channels (two channels, 387 and 607-nm wavelength in response to N2 532-nm excitation and one channel,
407-nm in response to water vapor 532-nm excitation)

– 2 depolarization channels (355 and 532 nm wavelength).

The lidar is then able to retrieve the aerosol profile of the particles at the three fundamental harmonics of 355, 532 and 1064
nm.

b) *Sunphotometer*

Sun photometers are powerful passive remote sensing instruments that measure the optical and microphysical properties
integrated into the atmospheric column of aerosols. These instruments work by measuring the intensity of direct solar radiation
at different wavelengths, which allows us to estimate atmospheric aerosol loading, optical properties, and their size distribution.

In this study, we used data from the Aerosol Robotic Network (AERONET;Holben et al. (1998)) station deployed at the
North Campus of the UPC and located in conjunction with the lidar instrument (41.38N, 2.11E, 125 m above sea level).

c) *PM sensor*

The hourly concentrations measured in $\mu g\ m^{-3}$ of PM10, PM2.5, and PM1, corrected by gravimetry (Charron et al., 2004),
are taken with the commercially available GRIMM OPC (Optical Particle Counter), which is an instrument used to measure
the size and number of particles in the air. This measurement was obtained by continuously sampling the air and counting the
number of particles that passed through the instrument's optical sensor.

Data for this study are obtained from the observational site deployed at CID (Center d'Investigació i Desenvolupament)
CSIC (Consejo Superior de Investigación Cientificas) in Barcelona (43.38N, 2.11E, 77 m above sea level). All the measured
variables, with specifications and references, are reported in Table 1.



### 2.1.1   Networks of instruments

a) *EARLINET*

The European Aerosol Research Lidar Network, EARLINET, established in 2000, is the first coordinated aerosol lidar network whose key objective is to provide a comprehensive, quantitative, and statistically significant database on the spatial and temporal distribution of aerosols on a continental scale ((Bösenberg and Matthias, 2003); (Pappalardo et al., 2014)). Currently, the network includes 31 active stations distributed across Europe (Pappalardo et al., 2014). Lidar observations within the

network are performed on a regular schedule of one daytime measurement per week around 12 solar times, when the boundary layer is usually well developed, and two nighttime measurements per week, with low background light, to perform Raman extinction measurements. In addition to routine measurements, further observations are devoted to monitoring special events such as desert dust outbreaks, e.g. (Ansmann et al., 2003); (Mona et al., 2006); (Papayannis et al., 2008); (Guerrero-Rascado et al., 2008); (Mamouri et al., 2013); (Nisantzi et al., 2015), forest fires, e.g., (Müller et al., 2005); (Amiridis et al., 2009);

(Alados-Arboledas et al., 2011), (Nisantzi et al., 2014), photochemical smog (Carnuth et al., 2002) and volcanic eruptions, e.g. (Pappalardo et al., 2004); (Wang et al., 2008); (Mattis et al., 2010); (Ansmann et al., 2010); (Gross et al., 2012); (Papayannis et al., 2012); (Sicard et al., 2012); (Wiegner et al., 2012); (Navas-Guzmán et al., 2013); (Pappalardo et al., 2013). EARLINET started correlative measurements for the spaceborne lidars on board NASA's CALIPSO (Cloud-Aerosol Lidar and Infrared Pathfinder Satellite Observation) in June 2006 (Pappalardo et al., 2010) and ESA's AEOLUS in August 2018. EARLINET is

currently a key component of the ACTRIS infrastructure, which represents a big step toward better coordination of atmospheric observations in Europe toward the establishment of the European component of an Integrated Atmospheric Global System as part of GEOSS, the Global Earth Observation System of Systems (Lautenbacher, 2005). EARLINET is also a contributing network to the Global Aerosol Watch Program of the World Meteorological Organization.

The Barcelona lidar station has been a member of EARLINET since its inception in 2000. The Barcelona lidar group at

CommSensLab-UPC has been developing lidar instruments since 1993. The present system has just been updated with a new laser in June 2022. The general characteristics appear in (Rodríguez-Gómez et al., 2022). The system currently has 3 elastic channels (UV, VIS, IR), 2 Raman channels using the pure rotational Raman effect (UV, VIS), 2 depolarization channels (UV, VIS), and 1 water vapor channel. The implementation of a new broadband fluorescence channel is ongoing.

For this study, we used the level 2 backscattering coefficient (quality guaranteed) at 532 nm and 1064 nm. Measurements

are taken following the EARLINET measurement schedule, as shown in (Bösenberg and Matthias, 2003) and (D'Amico et al., 2015). The observational site is located on campus north of UPC at 41.38N and 2.12E, 125 m above sea level.

b) *AERONET*

The NASA Aerosol Robotic Network (AERONET) is a globally distributed network of ground-based sunphotometers operated by academic institutions and government agencies. These are commercially available sunphotometers manufactured

by Cimel Electronique (Holben et al., 1998). AERONET has become a critical tool for studying aerosol properties, as the network provides long-term high-quality measurements that can be used to validate satellite observations and improve our understanding of aerosol impacts on the environment. AERONET database is openly available to the public and is widely used



by researchers, government agencies, and other stakeholders around the world to inform policy decisions and advance scientific understanding of the Earth system. The success of AERONET can be attributed to the standardization of instrumentation,
measurement protocols, and data processing algorithms, which ensure the quality and consistency of the data collected by the network.

## 2.2 Retrieval Methods

### 2.2.1 Optical properties

a) *The atmospheric backscatter profile by Raman lidar*

The main advantage of Raman lidar is that it can independently retrieve vertically resolved backscatter and extinction coefficients without assuming an approximate range independent lidar ratio, i.e., the ratio between the aerosol extinction coefficient and aerosol backscatter coefficient, which shows very large variability for aerosols (20sr-120sr; (Ackermann, 1998)). Multiple studies (Ferrare et al., 2006; Reichardt et al., 2012) have demonstrated that the precision of the optically resolved vertical properties of the aerosol obtained from Raman lidar is significantly influenced by the solar background. Therefore, to improve
their accuracy, measurements are performed primarily at night.

In this study, we use the Raman lidar inversion algorithm of Ansmann et al. (1990) to independently retrieve the vertical profile of aerosol extinction and aerosol backscatter at the elastic wavelength $\lambda_0$ ($\lambda_0$=355, 532 nm), and hence the aerosol lidar ratio profile given the Raman backscattered lidar return at $\lambda_R$ ($\lambda_R$=387, 607 nm, respectively). Here $\lambda_R$ denotes the Raman-shifted wavelength associated with the elastic wavelength $\lambda_0$.

The lidar backscatter profiles have a vertical resolution of 60 meters and are averaged monthly and seasonally (see Section 3.1 and Section 3.2). However, it is important to note that the lidar database is not uniform across all years. Some periods have more measurements than others, which affects the level of uncertainty associated with the different averaged profiles. The seasonal Mann-Kendall test (see Section 2.3.2) accounts for these differences in measurement frequency by weighting the averaged profiles accordingly.

b) *Aerosol Optical Depth*

Another fundamental parameter linked to the aerosol optical properties, the aerosol optical depth (AOD), is a measure of the attenuation of light due to the presence of aerosols in the atmosphere, and can be retrieved both by lidar, by integrating the extinction lidar profile over the altitude, and by sunphotometer, by measuring the amount of direct sunlight that reaches the Earth's surface and comparing it to the amount of sunlight that would have reached the surface if there were no aerosols in the
atmosphere. To retrieve the AOD, a sunphotometer typically takes measurements of the solar irradiance at several wavelengths.

The measurements are then used to calculate the AOD by applying the Beer-Bouger-Lambert law (Beer, 1852), which describes the intensity of a laser beam propagating in an inhomogeneous medium (the atmosphere) in relation to the total extinction coefficient (i.e. scattering + absorption), and hence to the aerosol presence. In Table 1 are reported all the different




optical and microphysical variables measured by lidar, sunphotometer, and in situ sensors with relative specifications are
reported. The other variables are retrieved from the main variables in Table 1.

| Instrument | Measured Variables | Specs | Accessory References |
|---|---|---|---|
| Lidar | Backscatter profile ($m^{-1}sr^{-1}$) | Sicard et al. (2009); Kumar et al. (2011); Zenteno-Hernández et al. (2021) | Ansmann et al. (1990) |
| Lidar | Volume depolarization profile | Sicard et al. (2009); Kumar et al. (2011); Zenteno-Hernández et al. (2021) | Cairo et al. (1999); Freudenthaler et al. (2009) |
| Sunphotometer | Aerosol Optical Depth, Ångstrom Exponent | Holben et al. (1998) | Eck and Holben (1999); Dubovik et al. (2002) |
| Sensor | PM10, 2.5, 1 conc. ($\mu g\ m^{-3}$) | Reche et al. (2022) | Charron et al. (2004) |

**Table 1.** Remote sensing and in situ instruments and the main relative measured variables used for this study

### 2.2.2 Microphysical properties

a) *Color ratio*

When comparing backscattered signals from two different wavelengths, lidar can distinguish between coarse and fine modes
of aerosols, providing important information on their size (Vaughan, 2004; Liu et al., 2017). Coarse mode aerosols are generally
larger than 1 micrometer ($\mu$m) in diameter and include particles such as dust (Landi et al., 2021), pollen (Sicard et al., 2021)
and sea salt. These aerosols are typically emitted from natural sources, such as deserts, oceans, and vegetation, as well as
anthropogenic sources, such as construction sites and agriculture.

Fine-mode aerosols are generally smaller than 1 $\mu$m in diameter and include particles such as sulfate, nitrate, ammonium,
organic carbon, and black carbon. These aerosols are typically emitted from anthropogenic sources, such as fossil fuel com-
bustion, biomass burning, and industrial processes, as well as from natural sources such as volcanoes and wildfires.

The aerosol backscatter coefficient at a wavelength of 1064nm, obtained from the ACTRIS/EARLINET UPC lidar, is used
to determine the aerosol size together with the aerosol backscatter coefficient at 532 nm. Shorter wavelengths (such as 532 nm)
are more sensitive to the aerosol fine mode, whereas longer wavelengths (such as 1064 nm) are more sensitive to the aerosol
coarse mode. Therefore, the color ratio is often used to differentiate between the coarse and fine modes of aerosols. The color
ratio (CR) is defined at each altitude $z$ as the ratio between the backscatter coefficient at 1064 nm $\beta_{1064}(z)$ and the backscatter
coefficient at 532 nm $\beta_{532}(z)$ are the backscatter coefficients at 1064 and 532 nm at altitude $z$.

The color ratio $\chi$ is typically expected to fall between zero and one. In general, smaller values of the color ratio are indicative
of smaller particles, while larger values ($\chi \approx 1$) correspond to larger particles for which geometric optics can provide an adequate
approximation of the scattering process. As shown in (Haarig et al., 2018), CR values greater than 0.7 are related to dust aerosols
from the Sahara, while CR values lower than 0.5 indicate smaller particles related to anthropogenic emissions. Between these
values, it is likely to be a mixture of anthropogenic aerosols and dust aerosols

CR can be also retrieved from in situ sensors by taking the ratio between PM2.5 and PM10 concentrations as shown in (Fan
et al., 2021). Again, values close to 1 indicate coarse aerosol, while values lower than 0.5 indicate a predominance of fine
aerosol mode.



b) *Ångström exponent retrieval*

Likewise, the Ångström exponent (AE) can differentiate between coarse and fine mode aerosols. It is defined as the slope of the logarithm of the aerosol optical depth (AOD) at two different wavelengths (440-870nm in this study). The most common way to retrieve AE is through measurements of AOD at two or more wavelengths using remote sensing techniques such as sun photometry (Dubovik et al., 2002) or satellite-based instruments (Levy et al., 2010). The sunphotometer measures solar

irradiance at multiple wavelengths. The AOD is then calculated by comparing the measured solar irradiance with the modeled solar irradiance, assuming a clear sky. The Ångström exponent can be calculated from the AOD measurements at different wavelengths.

The Ångström exponent is an important indicator of the primary size range of columnar aerosols, as the AOD spectral curve is strongly associated with the particle size distribution (Dubovik et al., 2002). Typically, AE falls within the range of 1.3 to

1.7 in urban or continental sites, has lower values between 0 and 0.6 when there is desert dust, and ranges between 0 and 0.35 in the presence of marine particles, as shown in (Hess et al., 1998; Smirnov et al., 2002; Eck and Holben, 1999; Mazzola et al., 2010). In this study, AE is retrieved from 440-870 nm wavelengths.

| Aerosol type | AE (440-870 nm) | References |
|---|---|---|
| Urban and Continental sites | 1.3-1.7 | (Hess et al., 1998; Mazzola et al., 2010) |
| Desert Dust | 0-0.6 | (Smirnov et al., 2002; Eck and Holben, 1999) |
| Marine environment | 0-0.35 | (Smirnov et al., 2002; Eck and Holben, 1999) |

**Table 2.** Typical AE values for different types of aerosols

c) *Lidar volume depolarization ratio*

UPC lidar instrument is also equipped with a depolarization channel at 532 nm (Sicard et al., 2009; Kumar et al., 2011;

Zenteno-Hernández et al., 2021). The laser signal emitted is 100% linearly polarized in a direction. The depolarization channel is measuring the backscattered energy in a direction that is perpendicular to the laser polarization, and then the degree of depolarization is measured. In this study, we use the Level 2.0 volume depolarization product, which is the sum of the molecular and particle depolarization. Detailed information on the volume depolarization channel can be found in Kumar et al. (2011) and Zenteno-Hernández et al. (2021). More generally, lidar depolarization processes can be found in Cairo et al. (1999) and

Freudenthaler et al. (2009).

d) *Aerosol typing*

Through the lidar volume polarization channel, it is possible to distinguish between different types of aerosol. Dust particles have irregular shapes that cause them to scatter light in a highly depolarized manner. On the other hand, other types of



aerosol, such as water droplets or pollution particles, have more spherical shapes and cause less depolarization of scattered
light (Freudenthaler et al., 2009; Lolli et al., 2011; Haarig et al., 2018).

Combustion products and anthropogenic aerosols, in general, found mainly in an urban environment or large metropolitan
areas, are predominant spherical (Volume Depolarization < 10%), whereas dust particles and ash from volcanic eruptions
(especially fresh) show irregular shapes, with a much higher volume depolarization value, as shown in (Haarig et al., 2018).
Volume depolarization is defined as the ratio between the total cross-backscatter coefficient and the parallel total backscatter
coefficient at 532nm.

    e) *Fine and Coarse mode and implication aerosol microphysics*

Using two independent lidar measurements at two different wavelengths allows for the differentiation between coarse and
fine aerosols, providing important information on aerosol microphysics (Mamouri and Ansmann, 2014). This is because the
size of the aerosol particles affects the way they interact with light at different wavelengths. The backscattered signal by fine
particles is more sensible at 532nm than at 1064nm, whereas coarse particles are less wavelength selective, and the difference
in backscattering at the two wavelengths is much less with respect to smaller particulate.

## 2.3   Statistical Methods

### 2.3.1   Quality assurance screening

To guarantee the highest possible quality of the final products and, at the same time, to ensure the homogeneity of the data
from different types of lidar systems, the EARLINET network has implemented a rigorous quality assurance program that all
affiliated stations are expected to meet (D'Amico et al., 2015, 2016). The implementation of the quality assurance program has
been and still is one of the main activities of the network and is a prerequisite to establish and monitor over time the performance
of different Lidar systems that are part of the network. The Quality Control of EARLINET applies to both lidar instruments
and analysis algorithms levels. Without the requirements of the quality assurance program, it is not possible to submit data
to the data center. Data that are not in compliance with EARLINET standards are rejected already in the submission phase.
Automatic feedback is provided to the Data Originator reporting all the problems incurred for each rejected file, fostering the
prompt resubmission of the data. There are two types of quality control procedures: • Basic quality control (BQC): technical
quality control on the submitted product. They are executed to ensure that the product is compliant mainly from a technical
point of view with the defined standard. • Advanced quality control (AQC): a series of physical checks applied to the input
product. They are executed to assess the quality from a physical point of view of the product. If one or more basic quality
controls fail, the submitted product is rejected and detailed feedback is sent to the data source. If the product satisfies the basic
quality controls, it is sent in batches to the advanced quality controls. If the input product fails at least one advanced quality
control, the product is labeled as Level 1 otherwise Level 2. Level 1 data are intended as data that users should treat with caution
considering failed QC procedures. Information about the result of the QC procedure is reported in the file itself for complete





traceability. Level 2 data are, instead, data that are fully compliant with what a user can expect from the ACTRIS/EARLINET network and with data documentation.

### 2.3.2    Year-by-year and seasonal statistical significance

*Seasonal Mann-Kendall test*

The most suitable way to determine temporal trends in aerosol optical depth (AOD), Ångström exponent (AE), and PM
concentrations is through a rank-based non-parametric method, which does not require a specific statistical distribution and can handle missing values, negative or below detection limit values. In this study, the Mann-Kendall (MK) test is used to identify increasing or decreasing long-term monotonic trends. The Sen slope estimator, which is based on the median of slopes calculated from all possible data pairs, is then used to calculate the slope and confidence limits (95% confidence level in this study). Due to the presence of distinct seasonal patterns in a large number of time series, the modified seasonal MK test
(Hirsch et al., 1982) was consistently used to assess trends over 12 months. The MK test is intended for data that are not serially correlated, so the presence of autocorrelation in a time series can affect the results and lead to a higher chance of erroneously rejecting the null hypothesis of no trend. This is known as a type 1 error or false positive. We ran a test on the monthly averaged data without finding any significant correlation (not shown). We then analyze the long-term trends for both AOD and AE retrieved from AERONET and PM10, PM2.5, and PM1 from in situ observation.

Data are aggregated on a monthly average (see below), and seasonal Mann-Kendall (MK) with Sen slope is computed to establish statistically significant temporal trends. MK test takes into account seasonality. The Mann-Kendall test was first proposed as an extension of the Spearman rank (Yue et al., 2002) correlation test. Although Spearman's rho can identify linear relationships between two variables, the Mann-Kendall test can detect both linear and non-linear changes over time. Furthermore, unlike other trend tests, such as least squares regression or analysis of variance, which require large sample
sizes for accurate estimates, Mann-Kendall is suitable for small samples that are often encountered in environmental studies (Von Storch and Zwiers, 2002).

For this study, we applied the seasonal Mann-Kendall test in AERONET Version 3 Level 2 quality-checked AOD, AE observations (Giles et al., 2019) and monthly averaged data of Level 2 PM concentration (see Section 3.1). The temporal resolution of the unprocessed AOD, AE, PM, and lidar observations can be found in Table 3.

The seasonal MK test is considered better than other temporal trend tests because of its ability to handle tied data, which can occur frequently in environmental data sets. Additionally, the Mann-Kendall test has greater statistical power than other trend tests, meaning that it is more likely to detect a trend when one exists, and it has a lower Type I error rate, which reduces the chances of falsely detecting a trend when one does not exist.

Average daily AOD and AE values are obtained from the data processing of AERONET level 2 version 3 (Giles et al., 2019).
The standard error on the monthly averages is calculated by dividing the standard deviation by the square root of the averaged number of samples, as shown in Eq. 1:



| Variable | Unprocessed resolution | Level 3 |
|---|---|---|
| AOD | Daily Averages (Giles et al., 2019) | Monthly Averages |
| AE | Daily Averages (Giles et al., 2019) | Monthly Averages |
| PMs | Hourly Averages (Reche et al., 2022) | Monthly Averages |
| Integrated backscatter (0-3 km) | EARLINET Database (D'Amico et al., 2015, 2016) | Seas. Averages (each year) |
| Integrated backscatter (3-8 km) | EARLINET Database (D'Amico et al., 2015, 2016) | Seas. Averages (each year) |

**Table 3.** Unprocessed temporal resolution of the data and new Level 3 used to determine long-term temporal trends

$$\sigma_e = \frac{\sigma}{\sqrt{n}} \tag{1}$$

where $\sigma$ is the standard deviation from the $n$ measures.

*Seasonal Variability*

We also checked the seasonal variability of AOD, AE, PM concentrations, and lidar backscatter profiles by averaging the variables and lidar profiles over four seasons, namely December-January-February (DJF), March-April-May (MAM), June-July-August (JJA), and September-October-November (SON). The unprocessed temporal resolutions of the concentrations of AOD, AE, PM are reported in Table 3, while the single-calculus chain (SCC) lidar profiles are quality-assured at Level 2, as reported in Section 2.3.1.

The vertically resolved lidar profiles for backscatter are seasonally averaged strictly following the ACTRIS/EARLINET schedule. Out-of-schedule measurements are ignored to avoid any possible bias due to particular conditions, i.e., biomass burning or dust outbreak.

       AOD and AE, even if they are not normally distributed (see Figure 8), if we take more than 30 samples, AOD and AE are approximately normally distributed. Again, the error in mean can be computed by dividing the standard deviation by the square

root of the averaged samples.

## 3    Results and Discussion

Lidar observations, due to the complexity of the instrument and the EARLINET measurement schedule, are much less dense with respect to PM10, PM2.5, PM1 and AERONET AOD and Ångstrom exponent measurements.

### 3.1    Long-term aerosol-load concentration trends (2004-2020)

As shown in Figure 1, seasonal MK test with Sen's slope on the concentrations of PM10, PM2.5, and PM1 at the surface level has decreased since 2004 with a statistically significant slope of -1.07 (CI 95% -1.31 -0.82), -0.73 (CI 95% -0.86 -0.59) and -0.68 (CI 95% -0.81 -0.59) $\mu$ g $m^{-3}$ per year, respectively. This corresponds to a reduction of 11.5, 9.5 and 8.5 $\mu$g $m^{-3}$ in 17





years, which corresponds to a decrease in the concentration of PM10, PM2.5, and PM1 by approximately 54%, 61% and 68% (95% confidence level).

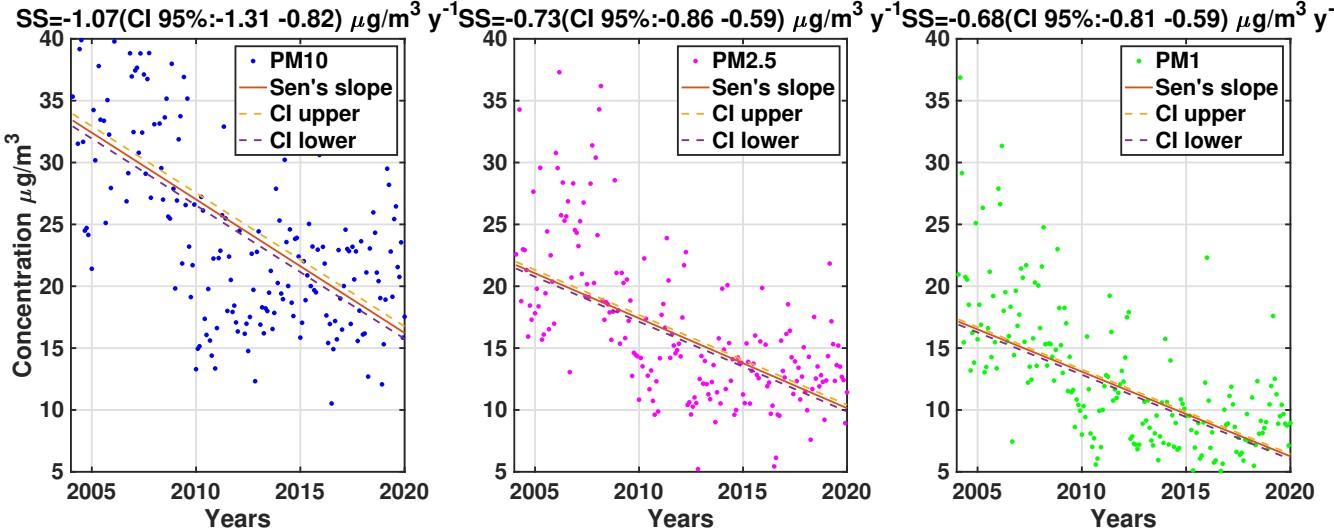

**Figure 1.** Plot of seasonal MK test for monthly averages of PM10, PM2.5, and PM1. A trend is evident at a confidence level 95% with a Sen slope of -1.07 (CI 95% -1.31 -0.82), -0.73 (CI 95% -0.86 -0.59) and -0.68 (CI 95% -0.81 -0.59) $\mu g\ m^{-3}\ y^{-1}$ respectively. These values correspond to a decrease of 54%, 61%, and 68% over 17 years.

This is an important result that confirms that the reduction policies implemented in recent decades were effective in improving air quality. In fact, PM1 is much less related to natural pollution episodes compared to PM10 and PM2.5 concentrations because fossil fuel combustion products are much smaller in size than dust, ash and marine salt outbreaks. However, reduction policies are ineffective in reducing the effects of natural and/or biogenic aerosol outbreaks, such as dust, marine salt, and volcanic eruptions.





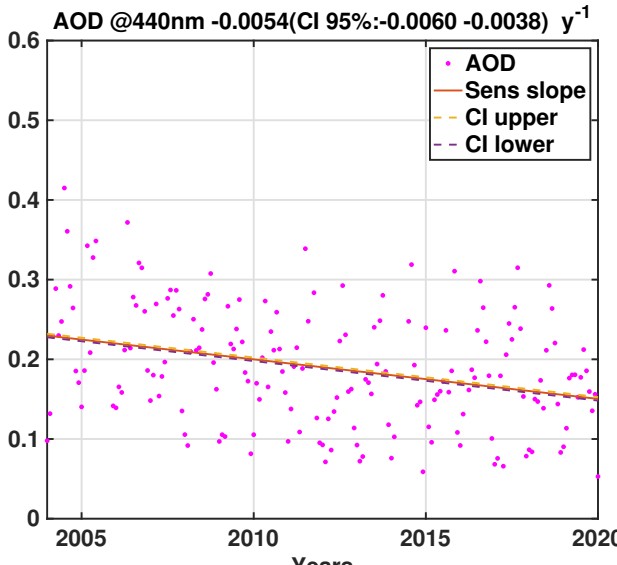
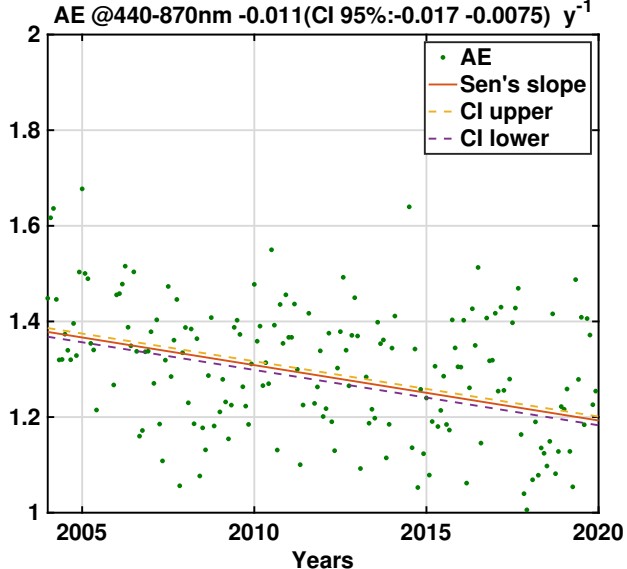

**Figure 2.** Plot of Seasonal MK test for AOD monthly averages at 440nm and Angstrom exponent from AERONET data. A trend is evident at a confidence level of 95% with a Sen slope of -0.0054 (CI 95% -0.0060 -0.0036) for the AOD and -0.011 (CI 95%:-0.017 -0.0075) for Ångstrom exponent.

The seasonal MK test with Sen's slope applied to monthly AOD averages at 440 nm showed a drop of 37% over 17 years (-0.0054 y-1; CI 95% -0.0060 -0.0038). The decrease in AOD is smaller than the drop in PM because AOD is a columnar value that includes the aerosol layers of the upper troposphere, usually advected from distant sources. For the Ångstrom exponent (AE), the drop in 17 years is 13% (-0.011 y-1; CI 95% -0.017 -0.0075). The drop in the ngström exponent confirms again that the reductions in emission policy were effective, since lower values of AE are related to larger aerosol particles, which in turn

are linked to natural emissions such as dust, ash, and marine salt.

     Lidar observations are fewer in number compared to those of AOD, AE, and PM because the instrument is much more complex. In this analysis, we apply the seasonal MK test to two parts of the yearly averaged backscatter profiles by season at 532 nm. The lower part of the backscatter, from the ground to the top of the boundary layer (0 km-3 km), and the free troposphere (3 km to 8 km) from 2004 to 2020.





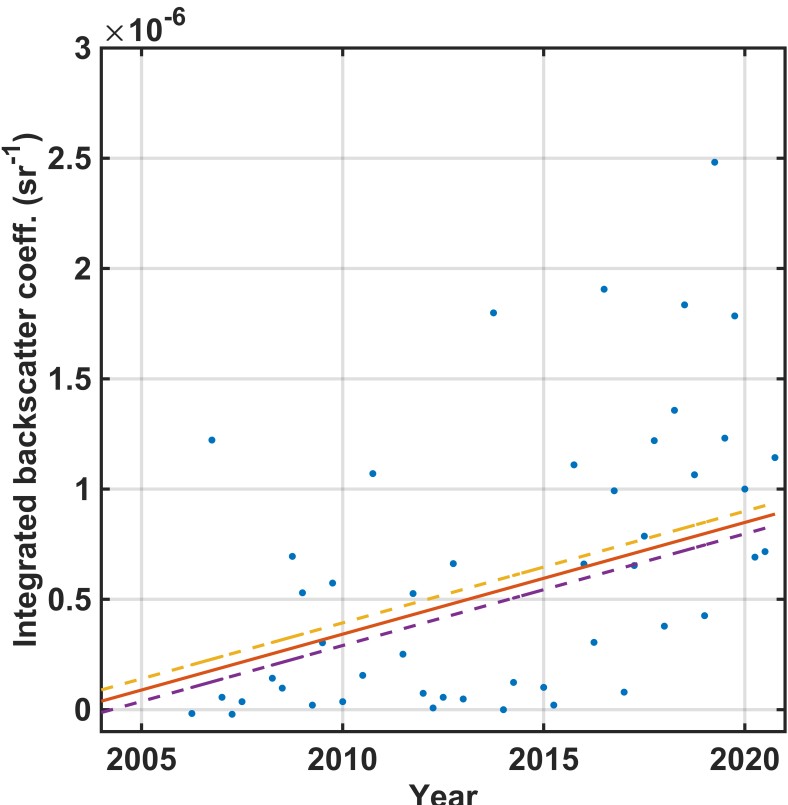

**Figure 3.** The MK test was applied to the averaged integrated backscatter profiles averaged by season (for each year) from 3 km to 8 km. The Sen slope is 5.48 10-8 $sr^{-1}\ y^{-1}$ (CI 95%: 2.48 10-8 $sr^{-1}\ y^{-1}$ - 7.24 10-8 $sr^{-1}\ y^{-1}$)

In the boundary layer (0 km-3 km), there is no significant trend at the 95% level, while in the free troposphere (3 km - 8 km) the average backscatter coefficient increases at a rate of 5.48 10e-8 $sr^{-1}y^{-1}$ (CI 95% 2.48 10-8 $sr^{-1}y^{-1}$ - 7.24 10e-8 $sr^{-1}y^{-1}$). For this tiny increase, we can then speculate that advected aerosol layers from distant sources, for example, dust outbreaks and biomass burning, have increased in the last 15 years. This speculation is also supported by (Evan et al., 2016). The same trend is not reflected in the boundary layer. This means that the reduction seen from sunphotometer and PM sensors

is mostly in the first hundreds meters where the lidar is blind or partially blind due to the overlap function.

### 3.2   Seasonal climatological analysis on long-term lidar observations

Lidar instruments have higher data collection and processing demands compared to in situ sensors for measuring PM and sunphotometers. Active remote sensing optical devices require more power, calibration, and maintenance to operate correctly, which can lead to data gaps. Additionally, lidar measurements are highly sensitive to atmospheric conditions, such as cloud

cover, which can affect their ability to detect aerosols on cloudy days. As a result, lidar observations may be less homogeneous, and some months/years may have more data than others. Consequently, it is very difficult to assess long-term trends with an in-



homogeneous database. However, it is possible to quantitatively assess the seasonal variability of the vertically resolved optical properties, which is crucial, particularly from a climatological standpoint. To achieve this result, we considered four seasons: December-January-February (DJF), March-April-May (MAM), June-July-August (JJA), and September-October-November

(SON). We obtained four averaged profiles, each representing the atmospheric aerosol load for each season. Profiles are obtained by taking the median average of the level 2 backscattering atmospheric profile. We used the median instead of the mean to eliminate the effects of outliers, for example, occasional aerosol plumes caused by land clearing activities by local landowners. From 2004 to 2020, lidar quality-assured aerosol backscatter profiles are 2630, both at 532 and 1064 nm. Due to cloud coverage and adverse meteorological conditions, median backscatters are obtained for the fall and winter seasons, averaging a

lower number of profiles compared to spring and summer, as shown in Table 4.

| Season | Averaged lidar profiles |
|--------|-------------------------|
| DJF    | 338                     |
| MAM    | 981                     |
| JJA    | 935                     |
| SON    | 378                     |

**Table 4.** Number of averaged lidar profiles by season 2004-2020).

The median average profiles are shown in Figure 4 with relative accuracy. At first glance, the summer aerosol layer, after a sharp decay in the first km, reaches an altitude of 5 km, whereas the other backscatter profiles during spring, fall, and winter exhibit a more pronounced exponential decay. During winter, the constant sharp drop of up to 1.7 km suggests aerosol of local origin, whereas the remaining profiles indicate a higher boundary layer height or upper air aerosol transport (max during

summer months).



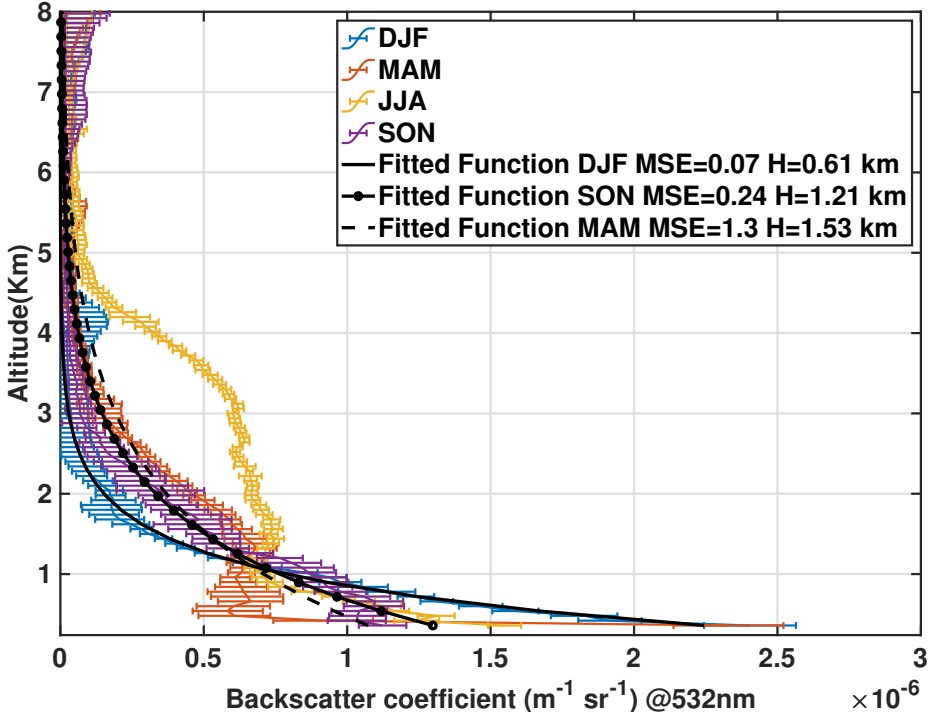

**Figure 4.** Vertically resolved median backscattering profiles for DJF, MAM, JJA, and SON. The error bars are obtained by computing the standard error (value $\pm 1\ \sigma_e$; Eq. 1). The spatial resolution is constant and fixed at 60 m. Black curves: Scale Height Retrieval and Relative Mean Square Error Fitting the backscatter profiles through a monoexponential function]

To better characterize the behavior of the backscatter profile, we fit the vertically resolved profiles of Figure 4 with an exponential function:

$$f(x) = A * exp(-\frac{x}{H}) \qquad (2)$$

where H is the scale factor indicated in our case at which altitude the backscattering drops by 63% (1/e). The fitted profiles
are depicted in Figure 4. The Mean Square Error (MSE) is very small for DJF, the plot showing an almost perfect fit and a scale height H=0.61 km. JJA is not represented because the curve cannot be fitted by an exponential function.

The Iberian Peninsula is also prone to dust outbreaks, especially during certain periods of the year (Sicard et al., 2011, 2012). To assess dust outbreaks, we analyzed the median-averaged depolarization ratio profiles, since dust particles strongly depolarize the lidar signal. For this reason, in Figure 5, the seasonal backscattering coefficient is paired with the seasonal vertically
resolved volume depolarization profile at 532nm, still obtained from the EARLINET/ACTRIS Barcelona station.

It is important to emphasize that backscatter measurements retrieve optical properties of aerosols, while volume depolarization is related to aerosol microphysical properties. In fact, a peak in the volume depolarization channel indicates the altitude at which the most depolarizing aerosols are found, which is not necessarily related to their concentration.



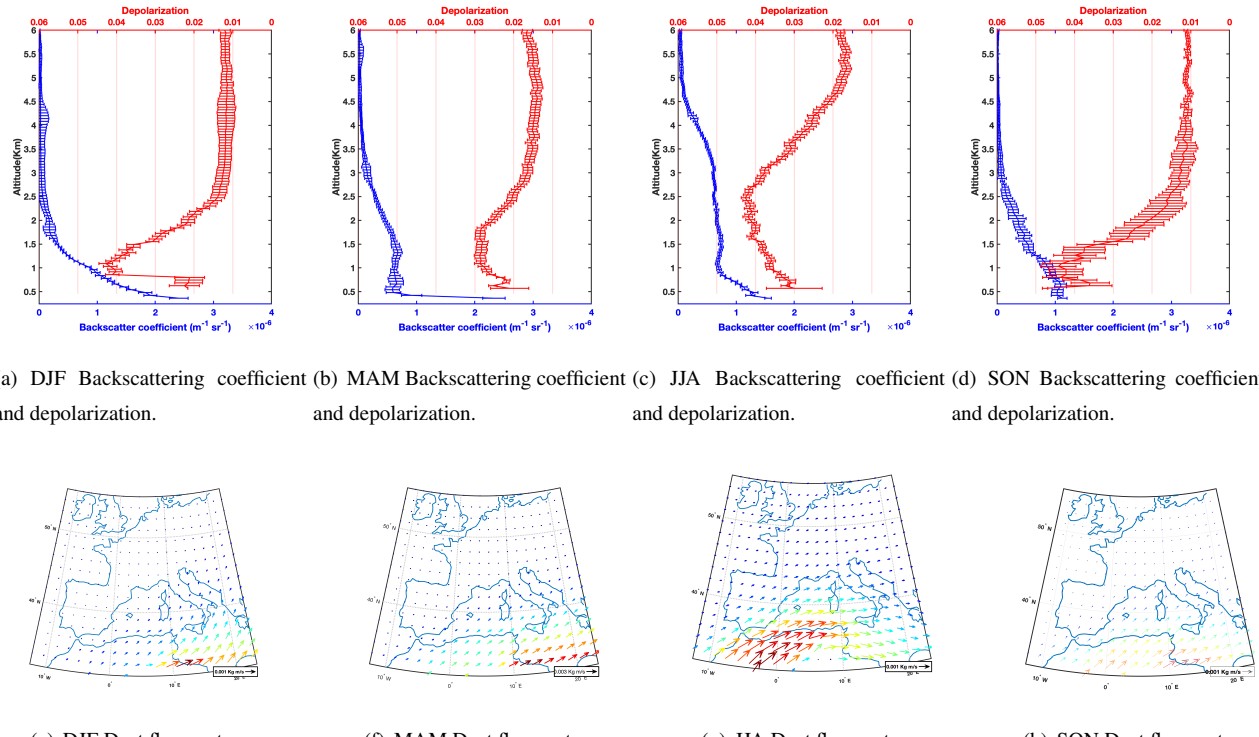

(a) DJF Backscattering coefficient and depolarization.   (b) MAM Backscattering coefficient and depolarization.   (c) JJA Backscattering coefficient and depolarization.   (d) SON Backscattering coefficient and depolarization.

(e) DJF Dust flux vector.   (f) MAM Dust flux vector.   (g) JJA Dust flux vector.   (h) SON Dust flux vector.

**Figure 5.** DJF, MAM, JJA, SON backscatter and depolarization vertically-resolved profiles at 532 nm. The errorbars are obtained computing the standard error (value $\pm$ 1 $\sigma_e$; Eq. 1). The spatial resolution is constant and set at 60 m. Synoptic maps were obtained from Modern era retrospective analysis for research and applications (MERRA; Giovanni (2015)). They represent the columnar dust flux averaged by season from 2004 to 2020.

In Figure 5, the vertically resolved volume depolarization profiles (in red) show significant variability that strongly depends
on the season. During winter (Figure 5(a)), there is a sharp volume depolarization peak at around 1 km, while during summer (Figure 5(c)), the curve is mostly flat from the ground up to 2.5 km, where the depolarization reaches a maximum. During spring and fall (Figures 5(b) and 5(d)), the peak is not as sharp as in winter and reaches an altitude of up to 1.5-2 km before starting to decay exponentially. The analysis showed that dust outbreaks are possible during all seasons, but there are fundamental differences.

During winter (DJF), long-range transport of aerosol layers is mostly floating at the top of the boundary layer, without long-range transport in the upper air at higher altitudes. It is also unlikely that dust will be found within the boundary layer. For this reason, during winter, the aerosols in the boundary layer are mainly local in origin and are not influenced by long-range transport. During summer (JJA), the depolarization curve is almost flat, indicating that dust aerosol layers can also be found within the boundary layer. The depolarization peak is found at an altitude of 2.5 km, followed by a sharper drop. The backscatter
coefficient (blue) shows the aerosols present up to an altitude of 5 Km. We speculate that long-range biomass-burning aerosol layers are advected toward the region. In fact, biomass burning is present in large part during the summer months, as shown





by (Ancellet et al., 2016). This type of aerosol does not exhibit significant volume depolarization, so it remains undetected by depolarization, but can be detected through backscatter. During the fall, the volume depolarization curve peaks around 1.3 km. The curve is not as sharp as in winter, indicating a possible intrusion of Saharian dust into the boundary layer. During spring,
although still probable, the volume depolarization values are 40% lower.

Figures 5(e), 5(f), 5(g), 5(h) show the seasonal columnar averaged dust flux from 2004 to 2020 obtained from NASA Geospatial Interactive Online Visualization ANd aNalysis Infrastructure, GIOVANNI Giovanni (2015). As we can observe, the dust flux over Barcelona is stronger during summer (JJA), while it is weaker during winter (DJF). Lidar observations also confirm the results found by (Marinou et al., 2017) using NASA Cloud-Aerosol Lidar and Infrared Pathfinder Satellite
Observations (CALIPSO) , where, even if the seasons are offset by a month, show that for Barcelona, latitude dust can be found up to 5 km in summer and 1 km in winter.

In summary, long-range dust outbreaks are likely to occur throughout the year. Volume depolarization values are lower in spring, indicating aged dust or mixing with other aerosol species, such as biomass burning or pollen(Sicard et al., 2021). During winter, the dust aerosol layer floats mostly at the top of the boundary layer, whereas in other seasons it can also be found inside.
Most of the dust can be found at 2.5 km during summer (up to 5 km), 1 km during winter, and 1 to 1.5 km during spring and fall. Several studies corroborated this speculation, and a reanalysis of seasonal maps (DJF, MAM, JJA, SON) of dust loading across Europe by the Barcelona Supercomputing Center (BSC) Dream model can be found in (Basart et al., 2012).

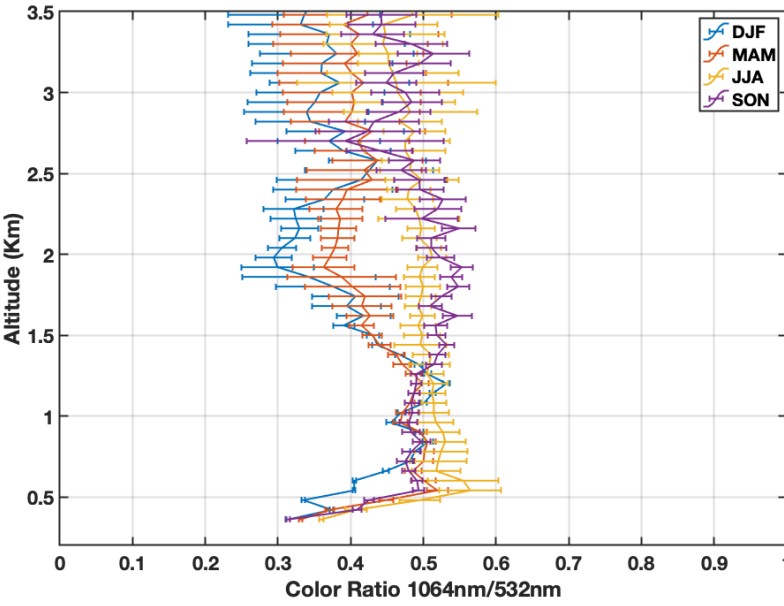

**Figure 6.** Color ratio for the different seasons obtained taking the ratio between the backscatter coefficient at 1064 nm and at 532 nm. Error bars are calculated considering the standard error (Eq. 1.)



### 3.2.1 Fine and Coarse mode

The vertically resolved color ratio accuracy is too low at 2.8 km, reaching errors of 100%. The seasonal signals can then be
exploited up to that altitude. Between 1 km and 2.8 km, the curves are well spread. CR profiles obtained through the median
averages indicate that the fine aerosol mode outweighs the coarse mode (resulting in fewer dust outbreaks). During DJF, a value
greater than 0.5 can be observed at approximately 1 km, indicating the presence of a potential aerosol layer floating above the
boundary layer, as previously observed from volume depolarization profiles. JJA and SON exhibit higher values compared to
DJF and MAM, supporting the findings from the volume depolarization profiles for that altitude range.

To perform a more in-depth analysis, we analyzed the seasonal averages of the Ångstrom exponent (microphysical variable)
and AOD (optical variable) retrievals from the AERONET sunphotometer (columnar variables) as well as the PM10, PM2.5,
and PM1 concentrations measured by in situ sensors co-located with the lidar and sunphotometer. The seasonal mean and
standard deviation values obtained from the PM concentration measurements are summarized in Table 5, while Figure 7 shows
all statistical values. PM10 concentrations shown in Figure 7 do not show any significant seasonal trend, as the mean values
remain relatively constant throughout the year. However, for PM2.5, as illustrated in Figure 7, the concentrations drop by
almost 20% during fall compared to winter. This trend is even more pronounced for PM1 (Figure 7), with a difference of 30%
between summer and winter concentrations. Lower concentrations during summer and fall are associated with higher mixing
layer height and convection. Both PM2.5 and PM1 concentrations exhibit greater variability during summer, as shown in Figure
7. In contrast, mean PM10 concentrations remain relatively stable throughout the year, indicating that heavier particles such
as marine salt maintain constant levels at the surface. This characteristic is unique to coastal metropolitan cities, compared to
continental regions such as the Po Valley, where PM10 concentrations vary significantly with seasons (Landi et al., 2021).

| Season | Group Count | PM10 | PM2.5 | PM1 | Std10 | Std2.5 | Std1 |
|--------|-------------|------|-------|-----|-------|--------|------|
| DJF | 36674 | 26.5 | 18.5 | 14.3 | 22.9 | 14.0 | 11.9 |
| MAM | 37536 | 27.1 | 16.8 | 12.2 | 21.5 | 11.7 | 10.2 |
| JJA | 37536 | 24.6 | 14.6 | 10.0 | 17.2 | 8.3 | 6.9 |
| SON | 37128 | 23.7 | 15.6 | 11.2 | 17.2 | 10.8 | 9.0 |

**Table 5.** In situ PM concentration ($\mu g/m^3$) sensor measurements seasonal averaged



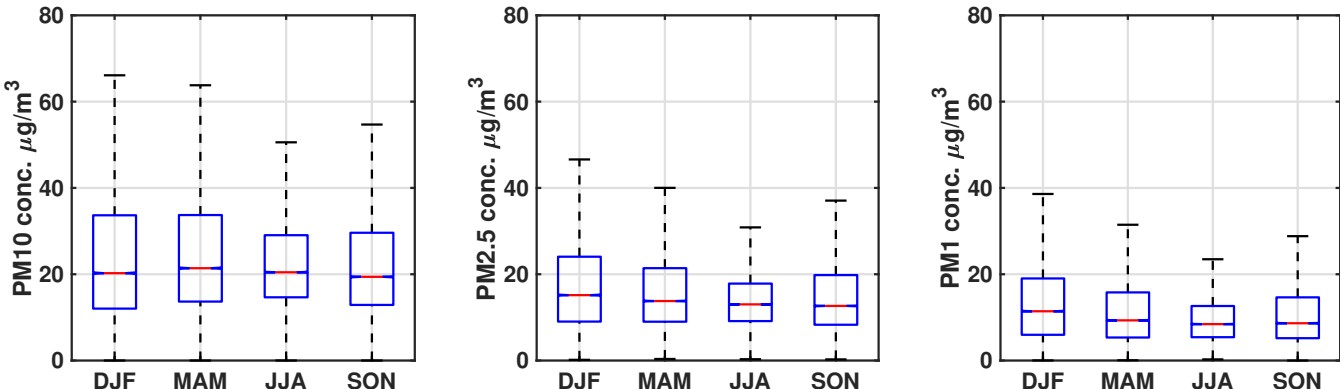

**Figure 7.** PM10, PM2.5, and PM1 seasonal in-situ concentrations. The bottom and top of each box are the 25th and 75th percentiles of the sample, respectively. The red line at the center of each box represents the median of the sample. Whiskers are lines that extend above and under each box. Observations beyond the whisker length are marked as outliers (in red). Seasonal variability is more pronounced during summer for fine particles, with higher concentrations during winter. Coarse aerosols show a lower seasonal variability

Figure 9(a) displays the relative frequency distributions of AOD at the 440 nm wavelength. The results show that approximately 57% of the AOD values are below 0.2 and approximately 75% are below 0.3, while only 5% are above 0.4. Figure 8 shows the statistical parameters of the seasonal AOD for winter, spring, summer, and fall. The highest average value is observed during summer, while the lowest value is observed during winter. The highest variability in the data is observed during spring, which may be due to the influence of pollen (Sicard et al., 2021).

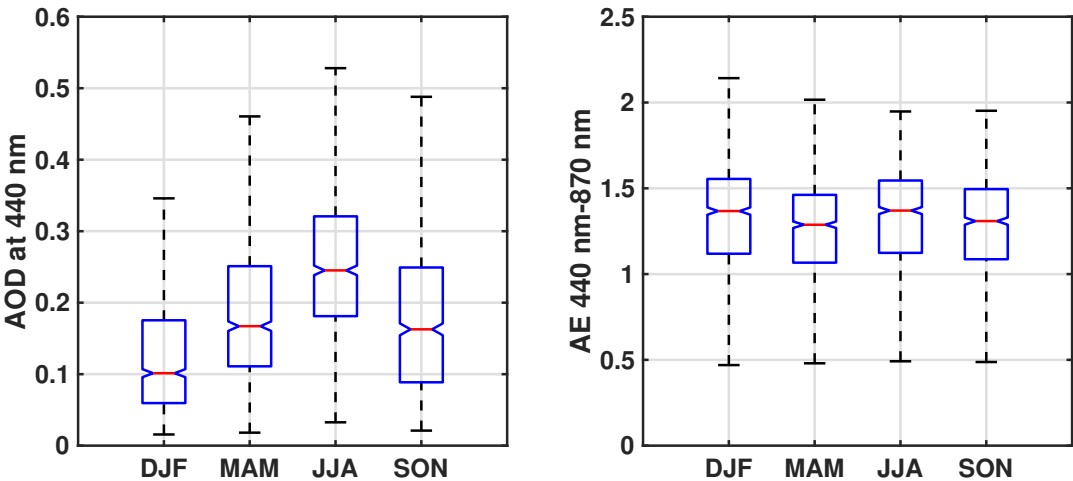

**Figure 8.** AERONET AOD and Ångstrom exponent seasonal boxplot. The bottom and top of each box are the 25th and 75th percentiles of the sample, respectively. The distance between the bottoms and tops of each box is the interquartile range. The red line in the center of each box represents the median of the sample. Whiskers are lines that extend above and below each box and correspond to the 95.4 percentile.



| Season | Group Count | AOD | Ångstr. Exponent | StdAOD | StdÅng |
|--------|-------------|-----|------------------|--------|--------|
| DJF | 1044 | 0.13 | 1.31 | 0.11 | 0.34 |
| MAM | 1098 | 0.20 | 1.25 | 0.12 | 0.32 |
| JJA | 1077 | 0.26 | 1.30 | 0.11 | 0.34 |
| SON | 927 | 0.18 | 1.27 | 0.12 | 0.30 |

**Table 6.** AERONET seasonal mean values and std for AOD and Ångstrom coefficient

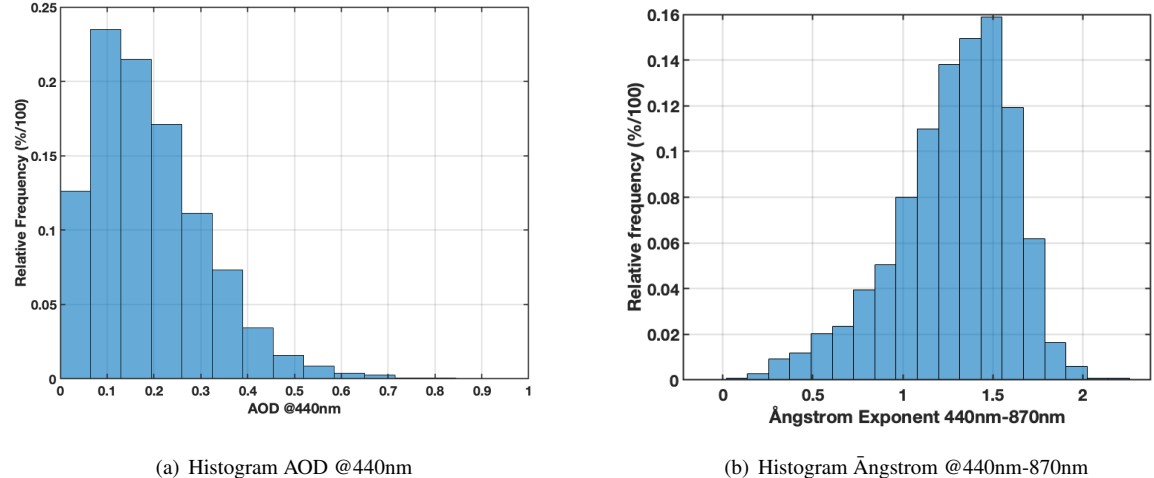

(a) Histogram AOD @440nm

(b) Histogram Āngstrom @440nm-870nm

**Figure 9.** Relative frequency histogram of AOD (440 nm) and AE (440 nm-870 nm) from the UPC Barcelona AERONET station

Figure 9(b) shows a relative frequency histogram of AE (440–870 nm), which exhibits an asymmetrical shape with a left tail that extends far.

Most of the values, which represent approximately 55% of the cases, were between 1.2 and 1.6, consistent with the results of other urban/industrial areas. The presence of low AE values, less than 0.7, in approximately 8% of the data indicates that desert dust transport events in the Catalonia region are not uncommon. Intermediate values of AE are probably due to mixed cases, where a thin layer of desert dust is present at altitudes above the boundary layer and contributes to the effects of extinction, combined with those caused by suspended particle loading in the lower troposphere.

The statistical parameters of AE are shown in Figure 8 for the whole period. There is no high variability throughout the season. Figure 8 confirms that long-range transported Saharan dust has the same chance of occurring during all seasons, with the AE almost constant throughout the seasons.



## 4  Conclusions and Future Perspectives

The present article discusses the impact of aerosols on human health, the environment and the climate in Barcelona, one of the largest metropolitan areas in the Mediterranean region. Aerosols are short-lived tiny particles suspended in the atmosphere and can come from both natural sources, such as volcano eruptions and dust outbreaks, and anthropogenic activities, such as burning fossil fuels. These particles can be carried by the wind from far away and can be found in high concentrations in certain areas. Breathing at high concentrations of aerosols can be harmful to human health, especially in vulnerable groups such as children and the elderly. It can cause respiratory problems, heart disease, and stroke. In contrast, the aerosols also modulate both the short-wave incoming solar radiation and the outgoing long-wave Earth radiation. In addition, aerosols influence cloud formation and lifetime and impact precipitation. The latest Intergovernmental Panel on Climate Change (IPCC) highlighted the need for aerosol research to reduce the current uncertainty. For this reason, in this study, we quantitatively evaluated changes in the optical and microphysical properties of aerosols in Barcelona during the last 17 years through the seasonal Mann-Kendall test that analyzes long-term temporal trends in the concentrations of retrieved AOD, the Ångström exponent and PM10, PM2.5 and PM1. The results highlight a sharp drop in PM concentration, AOD, and Ångström exponent from 2004 to 2020. These findings confirm that the emission reduction policies, implemented since 2004, were effective in improving air quality. From a climatological point of view, we also quantitatively assessed how the seasons influence the variability of the vertically resolved optical and microphysical properties of aerosols by lidar measurements obtained from the permanent station of the University Politecnica de Catalunya of the ACTRIS research infrastructure. We obtained four averaged profiles over 17 years of measurements, each representing the vertically resolved atmospheric aerosol backscatter profiles of each season.

The summer aerosol backscatter profile, averaged during June, July, and August, shows an exponential decay in the first km but also a long tail up to 5 km. This means that aerosol loading during this season can be found up to that altitude. On the contrary, the winter backscatter profile, which is obtained averaging the lidar observations in December, January, and February, shows a sharp exponential decay, with aerosols absent above 2.5 km. Both the spring aerosol backscatter profile, obtained averaging the lidar observations in March, April, and May, and the fall profile (averaged on September, October, and November) still show an exponential decay, but with a longer tail with detected aerosol presence up to 3 km in both cases. Spring and fall, and especially winter aerosol backscatter profiles, can be modeled by an exponential curve with a relative scale height H. In winter, for example, we have a scale height of H=0.61 km. This implies that 63% of the aerosol loading is capped below H.

Analyzing the lidar seasonal averaged volume depolarization profiles, which show higher values for non-spherical aerosols such as Saharan dust, and the color ratio, defined as the ratio between the backscatter coefficient at 1064 nm and the backscatter coefficient at 532 nm wavelengths, higher for larger aerosol particles, we found that in winter dust outbreaks are likely to float over the boundary layer at an altitude of 1 km, while for the other seasons, dust can also descend into the boundary layer, being the profiles smoother. During the summer, most of the dust is capped at 2.5 km. This altitude probably corresponds to the height of the center of the Saharan dust layers. In summary, dust outbreaks happen throughout the season, but in winter dust floats over the boundary layer, while in the other seasons it can be embedded into it.



It is also important to emphasize the differences between ground-based in situ measurements (PM concentrations) and columnar (AOD and AE) and vertically resolved lidar observation, i.e., lidar data show a net increase above 3 km over the analyzed period in contrast to the yearly drop at surface of PM concentrations. Seasonal analysis also shows differences in aerosols at different altitudes that are not detectable by analyzing only the concentration of PM on the surface. This study then
provides evidence that lidar observations are crucial to accurately evaluate aerosol dynamics. The ground-based observations do not capture the full picture.

This is the first study of this kind for the Iberian peninsula, and future research will evaluate whether the results found are confirmed in coastal and continental metropolitan regions of the Mediterranean, a climatic hotspot.

*Author contributions.* SL and TCL designed the study, SL, MS, AC performed the analysis, FA provided data for lidar, AA, CR, XQ provided
data for PM, and all of the authors assisted in interpreting the results and writing the paper.

*Competing interests.* At least one of the (co-)authors is a member of the editorial board of Atmospheric Chemistry and Physics



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
