# Peer review of "Climatological assessment of the vertically resolved optical and microphysical aerosol properties by lidar measurements, sunphotometer, and in-situ observations over 17 years at UPC Barcelona"

_EGUsphere, 2023_

## Author Comment (AC1)

Both VDR and backscatter profile for MAM, JJA MA (without May) and May. 1 bin = 60m

---

## Author Response (AR1)

Main changes to the document and responses to the reviewers: all changes are tracked in red in the new version.

Rev#1

Line 50: We have added references and further elaborated on the issue of emissions uncertainty, specifically highlighting the difference in emissions from top-down and bottom-up approaches and their alignment with LIDAR and AERONET observations, as suggested.

Line 165 and lines 346-348: We added a picture and a paragraph with the results for high intense and short episodes. We also found that the methodology is effective in detecting intense short-lived high pollution events, especially during the cold season. We found that there is an increase in intensity and frequency of Saharan dust outbreaks starting from 2016.

Line 190: We incorporated a discussion about the mixed nature of non-locally sourced aerosols, especially the smaller particles. The in-situ growth and core/shell mixing approaches consistent with AERONET have also been mentioned.

Line 200/201:. We have expanded our discussion to consider the possibility of aged anthropogenic or fire-based aerosols near the fine/coarse mode boundary.

Line 217: Based on your recommendation, we clarified this point and we added some more text.

Line 240: We have made necessary corrections for clarity.

Line 333: We have now included references to other research on long-range biomass burning observations, expanding the context and strengthening our claims.

Line 346-348: We re-analyzed our data focusing on the 95th percentile, as you suggested. The results from this analysis have enriched our findings and have been included in the revised version (see response for Line 165)

Figure 4 and Lines 375-384: We separated May data from MAM to compare it with March, April, and JJA. We checked if May is more similar to JJA or MAM. We found that May is closer to JJA. it aligns closer to summer months than spring. We added a paragraph about that.

Rev#2

1. All the number in PM10, PM2.5 and PM1 in the main text, tables and figures should be subscripted
   We changed them accordingly
2. What is Lidar detection blind zone?
   We added a paragraph to better explain the concept
3. Sens->Sen's in figure 2a
   We changed it accordingly
4. Brackets is not full in caption of table 4
   We changed it accordingly
5. Color bar is missed in figure 5
   We changed it accordingly
6. The present findings should be compared with previous studies.
   We added a paragraph in the discussion where we compare our results with previous studies.